# Total Hip Arthroplasty Using Imageless Computer-Assisted Navigation—2-Year Follow-Up of a Prospective Randomized Study

**DOI:** 10.3390/jcm9061620

**Published:** 2020-05-27

**Authors:** Richard Lass, Boris Olischar, Bernd Kubista, Thomas Waldhoer, Alexander Giurea, Reinhard Windhager

**Affiliations:** 1Department of Orthopedics and Trauma Surgery, Medical University of Vienna, Waehringer Guertel 18-20, 1090 Vienna, Austria; boris.olischar@gmail.com (B.O.); bernd.kubista@meduniwien.ac.at (B.K.); a.giurea@gmx.at (A.G.); reinhard.windhager@meduniwien.ac.at (R.W.); 2Department of Epidemiology, Center of Public Health, Medical University of Vienna, Kinderspitalgasse 15, 1090 Vienna, Austria; thomas.waldhoer@meduniwien.ac.at

**Keywords:** prospective randomized study, navigated total hip arthroplasty, accuracy assessment, cup placement, mid-term results

## Abstract

The purpose of this study is to compare computer-assisted to manual implantation-techniques in total hip arthroplasty (THA) and to find out if the computer-assisted surgery is able to improve the clinical and functional results and reduce the dislocation rate in short-terms after THA. We performed a concise minimum 2-year follow-up of the patient cohort of a prospective randomized study published in 2014 and evaluated if the higher implantation accuracy in the navigated group can be seen as an important determinant of success in total hip arthroplasty. Although a significant difference was found in mean postoperative acetabular component anteversion and in the outliers regarding inclination and anteversion (*p* < 0.05) between the computer-assisted and the manual-placed group, we could not find significant differences regarding clinical outcome or revision rates at 2-years follow-up. The implantation accuracy in the navigated group can be regarded as an important determinant of success in THA, although no significant differences in clinical outcome could be detected at short-term follow-up. Therefore, further long-term follow-up of our patient group is needed.

## 1. Introduction

Total hip arthroplasty (THA) has been described as one of the most successful orthopedic interventions for patients with degenerative hip diseases. An optimal selection and precise placement of the acetabular cup are two of the important factors for success of THA with regard to component migration [1] resulting in good mid- and long-term success. Malpositioning of the acetabular component in THA may result in complications like impingement [1], increased wear of the polyethylene (PE) liner [1,2], limited range of motion [1,2,3], joint dislocation [1,4], periprosthetic osteolysis [5], aseptic loosening of the prosthesis, and component migration [1]. These circumstances result in early revision arthroplasty [6,7,8,9].

Several studies regarding the optimal orientation of the acetabular component in THA have been published. Lewinnek et al. [10,11] proposed a “safe-zone“ for the cup orientation and recommended an anteversion angle of 15° ± 10° and an inclination angle of 40° ± 10°. He found out that cups positioned outside this zone had a considerably increased risk of dislocation. Posterior dislocation was observed by an angle below 5° of anteversion and anterior dislocation by an angle above 25°. 

In the past decade, acetabular cup positioning in THA has been based on the use of anatomical landmarks and freehand techniques depending on the experience of the orthopedic surgeon. Especially when using minimal invasive techniques, the use of computer-assisted surgery seems to be a good solution to the limited visibility of anatomical landmarks [12]. As conventional techniques have resulted in inaccurate cup inclination and anteversion placed outside the predefined safe-zone defined by Lewinnek [13,14], more reliable techniques and tools have been developed in order to prevent malpositioning of the implants [8] and to reduce the outliers [15,16,17,18,19]. In the last few years, the use of computer-assisted navigation systems in orthopedic surgery is steadily increasing in order to improve the accuracy of component implantation.

Lass et al. [20] performed a prospective randomized controlled study including 125 patients (62 patients in the navigation group and 63 patients in the conventional group) and compared the acetabular component placement using an imageless navigation system with the conventional manual (freehand) technique. The purpose of that prospective, randomized study was to compare the accuracy of cup positioning of different implantation techniques in THA and to find out if the computer-assisted surgery is able to improve the functional results and reduce the dislocation rate, wear, and the revision rates at a concise follow-up. 

We hypothesized that THA performed with computer-assisted surgery (CAS), that achieves an increased accuracy of acetabular cup position and a decreased number of cases outside the safe zone accurately defined by Lewinnek et al., would have better clinical outcomes and less revisions. This is an important requirement to improve patient-reported outcomes, reduce acetabular polyethylene wear, and increase survivorship in the long term.

## 2. Materials and Methods

### 2.1. Study Design and Patients

This study is a concise minimum 2-year follow-up of the prospective randomized study by Lass et al. published in 2014 [20], with regard to impingement of the prosthesis, joint dislocation, increased wear of the polyethylene liner due to uneven stress caused by malpositioning of the acetabular component, periprosthestic osteolysis, and aseptic loosening. We continued to collect and evaluate data during a period from August 2012 to September 2016 at a minimum of two years after primary THA. 

Originally, 130 patients were selected for the prospective, randomized, controlled study divided into two groups of 65 patients each. The study received ethical approval from the regional institutional review board (Ethics Committee of the Medical University of Vienna, IRB Number 013/2008). An informed consent was given by every patient. Five out of 130 randomly assigned patients were withdrawn from the study due to internal diseases which were too severe for an elective hip arthroplasty. Finally, 125 cases underwent THA (62 navigated and 63 conventional) during the period from February 2009 to August 2012 [20]. In our follow-up, 15 patients (12% of the initial study group) were not available for evaluation in the observation period from August 2012 to September 2016. The remaining 110 patients underwent a clinical and radiological examination (54 navigated and 56 conventional). The following parameters were collected pre- and postoperatively; age, gender, body mass index, surgical treated side, clinical and functional outcome using the Western Ontario and McMaster Universities Arthritis Index (WOMAC) [21], and Harris Hip Score (HHS) [22], and compared between the two groups.

### 2.2. Study Parameters

Patients were recommended for follow-up examinations at 6 weeks, at 3 and 6 months, and then once a year after surgery. For the recent study, we continued to collect and evaluate data for a minimum of two years. 

To investigate the clinical and functional outcome, we analyzed the Harris Hip and the WOMAC score at each follow-up. Radiographs (standard anteroposterior pelvic view in standing position, the central beam directed at the symphysis) were taken at each follow-up visit and were used to measure the anteversion and inclination angles and leg length discrepancy and compare them to the postoperative values after 6 weeks to detect any implant migration. The cup position was measured by two independent observers (orthopedic surgeons) using a special evaluation system (Impax^®^ EE R20 XIX, Agfa, Germany) without significant difference in their results. The inclination of the acetabular cup was measured in the standard anteroposterior pelvic radiograph as the angle produced by the cup axis and a horizontal line drawn along both ischia (Figure 1a.). The cup version was measured using the method of Widmer [23] (Figure 1b).

The determination and comparison of the postoperative acetabular cup position between the computer-assisted and freehand group was already performed at the radiological investigation in 2014 using CT-scans, because it represents the most contemporary measurement technique, with the ability to precisely reference anatomic landmarks. Due to the high exposure to radiation when using CT-scans, we did not use that for the regular follow-up examination at a minimum follow-up of 2 years.

The type of prosthesis used in this study was a cementless tapered rectangular titanium stem and a cementless titanium conical screw cup (Alloclassic® Zweymüller® SL, Alloclassic™ Zweymüller^™^ CSF, Zimmer, Inc., Warsaw, IN, USA). The computer-assisted surgery was performed using an imageless computer navigation system (Navitrack^®^, ORTHOsoft, Universal Imageless THR 2.0 Inc., Montreal, Canada). A modified transgluteal approach in supine position was performed in all patients [20]. In the navigated group, the cup position was measured during the reaming procedure and the insertion of the cup and recorded by the computer system. In the conventional surgery group, the acetabular components were placed manually, using only mechanical alignment guides, relying on the surgeon’s ability to estimate the cup orientation in relation to the patient’s position on the operating table. Our target acetabular component position was an inclination of 40° and an anteversion of 15° according to Lewinnek et al. [10,20].

The mean total surgery time for the computer-assisted implantation, including pin placement, was 122.3 min (range: 65 to 170), and for the freehand-implantation was 104.2 min (range: 50 to 165). The mean additional time needed for the computer-assisted procedure was 18.1 minutes in our study (*p* = 0.01).

### 2.3. Statistical Analysis

Statistical analysis was conducted by the local Institute of Medical Statistics. Descriptive statistics were performed to describe the research groups (mean and range). Two-tailed independent samples *t*-tests were undertaken for the comparison of continuous data, like age and BMI at baseline as well as Harris Hip and WOMAC scores at the time of 2-year follow-up of the two groups. To compare sets of categorical data, like gender, side of operation or revision rates with regard to implant, or navigation-related incidents, Levene’s test for equality of variances were performed. The radiological results were analyzed using a two-group (computer-assisted, freehand technique) comparison of the postoperative inclination and anteversion after 24 months using nonparametrical tests (Kolmogorov–Smirnov). *p*-values < 0.05 were considered as statistically significant. 

## 3. Results

### 3.1. Demographic Data

There were no significant differences in demographic data between the two groups (Table 1). In total, 15 patients (12%) were not available for the follow-up evaluation. Seven patients (6.4%) died during the two-year observation period, five (4.5%) in the conventional group, and two (1.8%) in the navigation group. Six patients (5.4%) moved to foreign countries and refused to come for the examination at the 2-year follow-up. We contacted them by phone to get information about their THA. All of them reported a superior to excellent satisfaction. One patient (0.9%), who had a poor state of health, could not attend the examination due to immobility after surgical treatment of the spine. No problem in relation to THA was reported in this case.

Finally, 110 cases were examined, including 69 female (63%) and 41 male (37%) patients. There were 56 patients in the conventional surgery group (31 female and 25 male) and 54 patients in the navigation group (38 female and 16 male), without any significant demographic differences (Table 1).

### 3.2. Revisions

Revision surgery was required in three cases (2.7%): two (1.8%) revisions in the navigated and one (0.9%) revision in the conventional group. The patient in the conventional group, suffering from a periprosthetic infection, had a very poor state of health. Therefore, the patient underwent parenteral antiobiotic treatment and surgical lavage, without removing the well-fixed implant. In the navigation group, we had two cases: One was a periprosthetic fracture caused by osteoporosis. The patient was treated with stem exchange and plate osteosynthesis. The other patient was a periprosthetic infection with a removal of the primary prosthesis and a two-stage revision surgery. Those cases were excluded from our follow-up study and were not considered in the data evaluation. There were no revisions due to mechanical failure.

### 3.3. Radiological Results

The values of inclination and anteversion were compared in both groups using postoperative CT-scans on the second postoperative day before weight-bearing and X-ray after 3, 6, 12, and 24 months. The results of postoperative CT scans were already presented in the previous paper by Lass et al. [20] as follows.

The mean inclination was 38.6° ± 3.6° (range, 32° to 50°) in the navigated group and 37.7° ± 5.2° (range: 27° to 50°) in the freehand-placement group. The mean anteversion was 19.5° ± 4.6° (range: 9° to 29°) in the navigated group and 17.3° ± 10.4° (range, 0° to 44°) in the conventional freehand-placement group. There was no significant difference between the computer-assisted group and the conventional group with regard to the mean inclination angles (*p* = 0.29), but a significant difference with regard to the mean anteversion angle (*p* = 0.007). There were no outliers for inclination in the computer-assisted hips, but five of 63 outliers (7.9%) in the conventional group (*p* = 0.02). There were six of 62 (9.7%) outliers for anteversion in the navigated group, and 23 of 63 (36.5%) in the conventional group (*p* < 0.05).

In the radiologic follow-up investigation at a minimum of 2 years we could find no mechanical failure due to loosening induced migration compared to the mentioned values (Figure 2 and Figure 3). 

### 3.4. Clinical Results

Harris Hip Score and WOMAC Score improved significantly comparing pre- and postoperative in the computer-assisted as well as in the freehand placement group. There were no significant differences between the groups in the follow up examination at a mean time of 1.5 years (range: 0.1–3.5 years), which was already presented in the previous publication [20]. 

We still could not find a significant difference for the clinical results as the HHS (p = 0.3) or the WOMAC Score (*p* = 0.6) between the computer-assisted group and the freehand placement group at a minimum follow-up of 2 years (mean: 4.5; range: 2.2–6.9 years) (Table 1).

## 4. Discussion

Based on the recent literature, inaccurate acetabular component placement in THA may increase the risk of dislocation, reduce the range of motion due to intraarticular impingement, and cause increased acetabular wear. Malpositioning may also lead to increased wear of the polyethylene liner, osteolysis, and loosening of the prosthesis. Recently, CAS for THA has been reestablished onto the market using robotic-assisted computer navigation. Lass et al. [20] demonstrated an improved accuracy in positioning of the acetabular component using computer-assisted surgery concerning the anteversion angle in THA. 

Our concise follow-up study of this prospective, randomized, controlled study could confirm the results at a minimum follow up of 2 years without any signs of loosening or migration in the study group, neither in the conventional nor in the navigated group. Although the radiographic follow-up was not the main purpose of this recent study, we could show that CAS improved the acetabular component position as defined by Lewinnek et al. [10]. This is in accordance to recent studies showing the superiority of computer-assisted implantation in terms of accuracy [6,7,9,11,16,17,18,24,25,26,27,28,29,30,31,32]. Comparing with the studies of Kalteis et al. [28] using imageless computer-assisted implantation and Parrette and Argenson [33], who both performed CT-based navigation to measure the operative angles, we found similar results in the navigated group, showing that the use of an imageless navigation system can improve cup positioning in total hip arthroplasty by reducing the percentage of outliers. In the control group, however, we obtained better results, because it was very important for the surgeons to make a correct evaluation of the anatomic references as the acetabular walls corresponding to the pelvic osseous points, as described by Sotereanos et al. [34], or the transverse ligament, in light of the correct use of the mechanical alignment guide. Orientation only based on visual assessment by the surgeon is often responsible for inaccurate placement. An orthopedic surgeon who does not routinely perform THAs could perhaps benefit more from the computer-assisted implantation of the THA, especially of the acetabular cup. Furthermore, when using minimal invasive techniques, computer-assisted implantation seems to be the solution to the limited visibility of anatomical landmarks [11,33].

Parrette et al. [35] suggested that CAS used for cup placement does not have any substantial advantage in function, wear rate, or survivorship at 10 years after THA. They point out that precision in THA can be achieved by experienced surgeons in routine THA. Dorr has identified the problem in the population of patients, who benefit from computer navigation. We have to keep in mind that computer technology has advanced well beyond where it was 10 years ago, and the challenge is to determine how to properly use computer technology for sophisticated preoperative planning [36]. Rittmeister et al. [37] reported for 500 THAs performed free hand; postoperative radiographs revealed that 19.8% of the cups were outside the safe zone for anteversion and 11.2% for abduction. Kennedy et al. reported an 11% prevalence of pelvic osteolysis when the mean angle of abduction of the acetabular component was reduced from 61.9° to 49.3° [1]. 

Concerning the recent literature, only few studies have been published on clinical results of THA using computer navigation, most of them performed as prospective non-randomized, comparative trials. So far, it has not been clearly shown whether the navigation-related implantation results in better clinical outcomes. 

To our knowledge, this current study, comparing imageless computer-assisted with freehand implantation of cementless total hip arthroplasties, is still including the largest prospective randomized sample size published at this time. 

The major limitation of our study is the short-term follow-up period (range: 2.2–6.9 years). The clinical differences, with regard to dislocation rate, range of motion, and pain as well as wear and aseptic loosening of the implants, between patients treated with navigation and those treated with freehand cup placement have to be evaluated at intermediate and long-term follow-up time periods in order to demonstrate potential benefits for hip navigation.

The main aim of this recent study was to find out if the computer-assisted surgery is able to improve the functional results and reduce the dislocation rate and wear at a minimum of 2-year follow-up, although we still could not find a significant difference for the clinical results as the HHS (*p* = 0.3) and the revision rates between the computer-assisted group and the freehand placement group. 

## 5. Conclusions

Based on the results of our present study, we conclude that imageless hip navigation increases the accuracy of acetabular component placement within the desired position and safe zone compared with that achieved with conventional freehand implantation methods, but there is still a lack of evidence to link this cup position improvement to substantial clinical improvements at short- and midterm follow-up. Clinical results have to be evaluated at long-term follow-up in prospective randomized studies in a representative study group to find significant differences between the groups. 

## Figures and Tables

**Figure 1 jcm-09-01620-f001:**
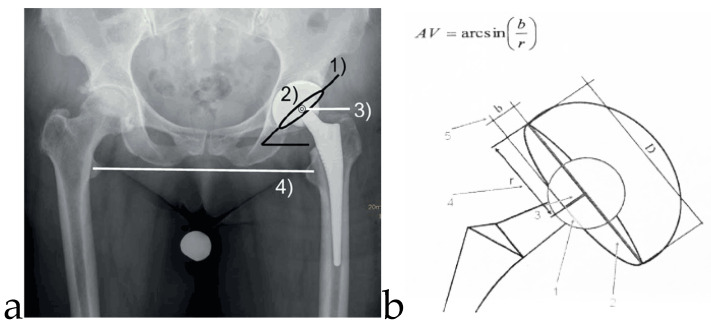
(**a**). Standard pelvic radiograph after total hip arthroplasty in standing position, the central beam directed at the symphysis. Cup inclination (1) and anteversion (2), rotational center of the hip (3), and leg length (4). (**b**). A simplified method to determine acetabular cup anteversion from plain radiographs (Widmer KH [23]). The short axis of the projected ellipse is measured and related to the total cross-section of the projected cup along the short axis. This relationship correlates with acetabular cup anteversion angles and represents an inverse sinus function.

**Figure 2 jcm-09-01620-f002:**
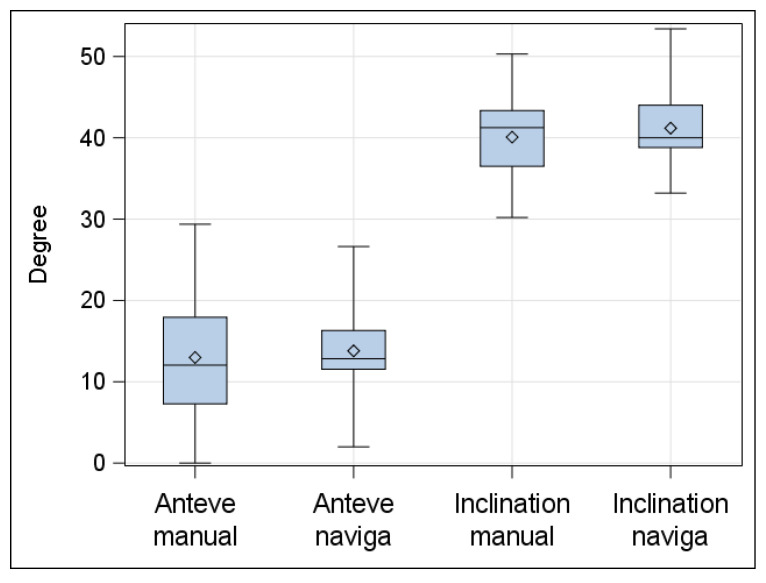
Postoperative mean anteversion and inclination (standard deviation) in the manual and navigated group after 24 months.

**Figure 3 jcm-09-01620-f003:**
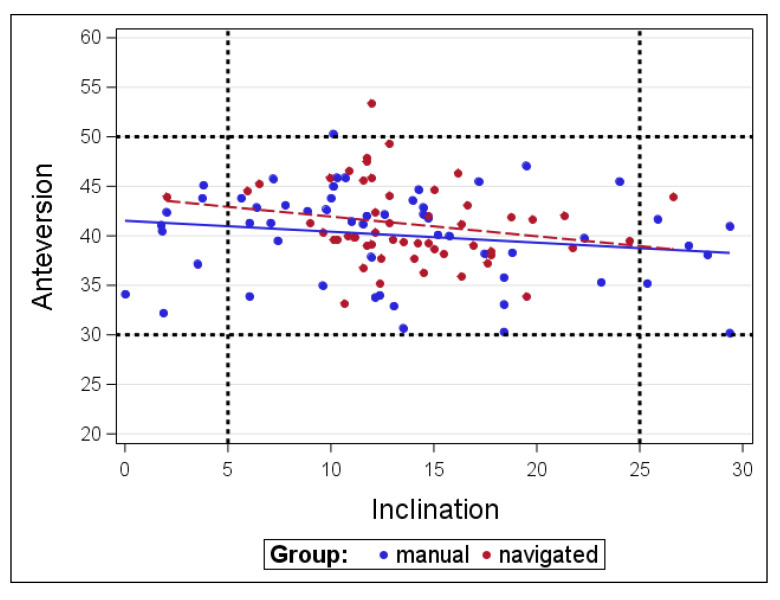
Distribution of anteversion and inclination angles measured in the postoperative x-ray after 24 months in the navigated and manual group.

**Table 1 jcm-09-01620-t001:** Demographic data and clinical results.

Type of Operation		Conventional			Navigated		
	n (%)	Mean	Range	n (%)	Mean	Range	Significance (*p*)
Gender; f/m	31/25 (55/45)			38/16 (70/30)			0.1
Side; r/l	35/21 (63/37)			33/21 (61/39)			0.88
Age (years)		79.5	40–100		77.2	40–98	0.15
BMI (kg/m^2^)		26.0	18–38		28.70	20–41	0.94
HHS preoperative		35.0	0–65		32.6	7–86	0.38
HHS postop. 6 weeks		90.2	62–100		90.9	61–100	0.72
HHS postop. 24 month		92.2	61–100		88.8	65–100	0.3
Leg length difference (mm)		4.4	0–20		2.7	0–15	0.09

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
