# Peer review of "Total Hip Arthroplasty Using Imageless Computer-Assisted Navigation—2-Year Follow-Up of a Prospective Randomized Study"

_jcm, 2020, doi:10.3390/jcm9061620_

Round 1

Reviewer 1 Report

The paper describes Minimum two-year results of a prospective randomized study comparing THA Implantation using either imageless computerassisted Navigation or manual implantation techniques.

The paper is well written, the introduction leads to the underlying hypothesis.

Line 91: Please state whether the classic Harris Hip Score or the modified Harris Hip Score was used. Please provide a reference for the HHS and the WOMAC.

Discussion: Though it was not a primary outcome of this study. Do you think that the surgical time was longer in the navigated group?

Congratulations to a well done and well written paper.

Author Response

Response to Reviewer 1 Comments

Dear Reviewer,

thank you for reviewing my manuscript, ID jcm-797960 „Total Hip Arthroplasty using Imageless Computer Assisted Navigation – 2 Year Follow-up of a Prospective Randomized Study” again, which was submitted for publication in the special issue “State-of-the-Art Research on Hip and Knee Arthroplasty” of the Journal of Clinical Medicine.

Concerns were addressed and a detailed response to your comments is added.

Point 1: Line 91: Please state whether the classic Harris Hip Score or the modified Harris Hip Score was used. Please provide a reference for the HHS and the WOMAC.

Response 1: The classic Harris Hip Score was used. The appropriate references for the HHS and the WOMAC Score were provided and added to the reference list.

(reference 21. and 22.) – (line 96)

Point 2: Discussion: Though it was not a primary outcome of this study. Do you think that the surgical time was longer in the navigated group?

Response 2: The extra time needed for calibration, registration and determination of the pointers and inserters was registered in the computer system and the additional surgical time was statistically evaluated and already mentioned in the paper published in 2014. Therefore, we didn’t want to point out that again, but because it is of interest for the reviewers and readers of the recent paper I added the following text to the Material and Methods of the submitted manuscript; (line 140-143)

The mean total surgery time for the computer-assisted implantation, including pin placement was 122.3 minutes (range, 65 to 170), for the freehand-implantation 104.2 minutes (range, 50 to 165). The mean additional time needed for the computer-assisted procedure was 18.1 minutes in our study (p=0.01).

Reviewer 2 Report

This paper is well written, with sound methods and well described results.

I would suggest including the limitations in the discussion, e.g. small sample size and short follow-up time.

Author Response

Response to Reviewer 2 Comments

Dear Reviewer,

thank you for reviewing my manuscript, ID jcm-797960 „Total Hip Arthroplasty using Imageless Computer Assisted Navigation – 2 Year Follow-up of a Prospective Randomized Study” again, which was submitted for publication in the special issue “State-of-the-Art Research on Hip and Knee Arthroplasty” of the Journal of Clinical Medicine.

Concerns were addressed and a detailed response to your comments is added.

Point 1: I would suggest including the limitations in the discussion, e.g. small sample size and short follow-up time.

Response 1: We included the limitations in the discussion as suggested; (line 263-267)

The major limitation of our study is the short-term follow-up period (range 2.2-6.9 years). The clinical differences, with regard to dislocation rate, range of motion, and pain as well as wear and aseptic loosening of the implants, between patients treated with navigation and those treated with freehand-cup-placement have to be evaluated at intermediate and long-term follow-up time-periods in order to demonstrate potential benefits for hip navigation.

Reviewer 3 Report

The result of this study is interesting to researchers in this field. However, some information should be added to the manuscript prior to publication. My comments are as follows.

  1. Figure 1b is not clear. Please make it clear.
  2. Line 104: Was there any difference in the evaluation result between the two observers? How about the reproducibility of the result?
  3. Line 193: Please indicate the reference that describes the previous result.
  4. Line 215: What kind of similar result? Please describe it in details.
  5. Line 242: I thought that it should not discuss about wear reduction within very short-term (2 years) follow-up. Do you have any comment on it?

Author Response

Response to Reviewer 3 Comments

Dear Reviewer,

thank you for reviewing my manuscript, ID jcm-797960 „Total Hip Arthroplasty using Imageless Computer Assisted Navigation – 2 Year Follow-up of a Prospective Randomized Study” again, which was submitted for publication in the special issue “State-of-the-Art Research on Hip and Knee Arthroplasty” of the Journal of Clinical Medicine.

Concerns were addressed and a detailed response to your comments is added.

Point 1: Figure 1b is not clear. Please make it clear.

Response 1: The legend of Figure 1b was completely rewritten to make it clear;
(line 110-111 and line 125-128)

A simplified method to determine acetabular cup anteversion from plain radiographs (Widmer KH [26]). The short axis of the projected ellipse is measured and related to the total cross-section of the projected cup along the short axis. This relationship correlates with acetabular cup anteversion angles and represents an inverse sinus function.

Point 2: Line 104: Was there any difference in the evaluation result between the two observers? How about the reproducibility of the result?

Response 2: There was no significant difference in their results. This information was added to the text; (line 108)

X-rays were performed at each follow-up visit and were used to detect any implant migration by comparing them to the postoperative values after 6 weeks. We didn’t use these measurements to evaluate the accuracy of the implantation technique, because reproducibility and the accuracy of the radiological evaluations based only on conventional plain x-rays must be regarded as difficult, particularly, because of problems in measuring the anteversion.

The following information was added to the manuscript; (line 112-118)

The determination and comparison of the postoperative acetabular cup position between the computer-assisted and freehand group was already performed at the radiological investigation in 2014 using CT-scans, because CT-scan represents the most contemporary measurement technique, with the ability to precisely reference anatomic landmarks. Due to the high exposure to radiation when using CT-scans we didn’t use that for the regular follow-up examination at a minimum follow-up of 2 years.

The results of postoperative CT-scans were already presented in the previous paper by Lass et al. and are mentioned in the Radiological Results of the recent paper.

Point 3: Line 193: Please indicate the reference that describes the previous result.

Response 3: The reference that describes the previous results is added to the text.
(line 211)

Point 4: Line 215: What kind of similar result? Please describe it in details.

Response 4; Both studies showed a significant reduction in variation of the position of the acetabular component using computer-assisted compared with conventional freehand arthroplasty and concluded that the use of an imageless navigation system can improve cup positioning in total hip arthroplasty by reducing the percentage of outliers.

We added that information to the manuscript; (line 233-234)

…, showing that the use of an imageless navigation system can improve cup positioning in total hip arthroplasty by reducing the percentage of outliers.

Point 5: Line 242: I thought that it should not discuss about wear reduction within very short-term (2 years) follow-up. Do you have any comment on it?

Response 5: I absolutely agree with you and added some information about that to the discussion; (line 263-267)

The clinical differences, with regard to dislocation rate, range of motion, and pain as well as wear and aseptic loosening of the implants, between patients treated with navigation and those treated with freehand-cup-placement have to be evaluated at intermediate and long-term follow-up time-periods in order to demonstrate potential benefits for hip navigation.

Round 2

Reviewer 3 Report

Thank you for your answers. It can be accepted now.